

# Evaluating the adaptive potential of the European eel: is the immunogenetic status recovering?

Miguel Baltazar-Soares[1], Seraina E. Bracamonte[1,2], Till Bayer[1], Frédéric J.J. Chain[3], Reinhold Hanel[4], Chris Harrod[5] and Christophe Eizaguirre[6]

[1] Evolutionary Ecology of Marine Fishes, GEOMAR Helmholtz Centre for Ocean Research Kiel, Kiel, Germany
[2] Leibniz-Institute of Freshwater Ecology and Inland Fisheries, Berlin, Germany
[3] Department of Biology, McGill University, Montréal Québec, Canada
[4] Thünen-Institute of Fisheries Ecology, Hamburg, Germany
[5] Universidad de Antofagasta, Instituto de Ciencias Naturales Alexander von Humboldt, Antofagasta, Chile
[6] School of Biological and Chemical Sciences, Queen Mary University of London, London, United Kingdom

Corresponding author
Miguel Baltazar-Soares,
msoares@geomar.de

## ABSTRACT

The recent increased integration of evolutionary theory into conservation programs has greatly improved our ability to protect endangered species. A common application of such theory links population dynamics and indices of genetic diversity, usually estimated from neutrally evolving markers. However, some studies have suggested that highly polymorphic adaptive genes, such as the immune genes of the Major Histocompatibility Complex (MHC), might be more sensitive to fluctuations in population dynamics. As such, the combination of neutrally- and adaptively-evolving genes may be informative in populations where reductions in abundance have been documented. The European eel (*Anguilla anguilla*) underwent a drastic and well-reported decline in abundance in the late 20th century and still displays low recruitment. Here we compared genetic diversity indices estimated from neutral (mitochondrial DNA and microsatellites) and adaptive markers (MHC) between two distinct generations of European eels. Our results revealed a clear discrepancy between signatures obtained for each class of markers. Although mtDNA and microsatellites showed no changes in diversity between the older and the younger generations, MHC diversity revealed a contemporary drop followed by a recent increase. Our results suggest ongoing gain of MHC genetic diversity resulting from the interplay between drift and selection and ultimately increasing the adaptive potential of the species.

# INTRODUCTION

Preserving natural biodiversity while allowing species to maintain their adaptive potential is a major challenge in modern conservation biology (*Frankham, Briscoe & Ballou, 2002*). Anthropogenic activities impact global ecosystems and reduce population sizes of species, whether by shrinking or fragmenting available habitats, overexploitation, or disruption

of population dynamics (*Allendorf et al., 2008*; *England, Luikart & Waples, 2010*; *Thomas et al., 2004*). Small populations are more vulnerable to environmental, demographic and genetic processes (*Keith et al., 2008*). Genetic factors are particularly important as they may not manifest immediately after population reduction but their effects persist in the population even if the population size recovers to otherwise sustainable levels, e.g., through bottleneck effects (*Spielman, Brook & Frankham, 2004*). Populations display complex evolutionary dynamics and evolutionary genetics offers ideal framework for conservation biologists to monitor population changes and viability (*Hendry et al., 2011*). Although genetic studies applied to wild populations of non-model species have largely focused on the analysis of neutrally evolving loci (see *McMahon, Teeling & Höglund, 2014*), conservation managers have expanded their toolbox to include screening of adaptive loci (*Hendry et al., 2011*). Such measures are required to fill knowledge gaps regarding species' evolutionary and adaptive potential (*Eizaguirre & Baltazar-Soares, 2014*).

The genes of the Major Histocompatibility Complex (MHC) have repeatedly been shown to be suitable candidates to evaluate the immune adaptive potential of endangered populations (*Sommer, 2005*; *Stiebens et al., 2013b*). This highly polymorphic (high heterozygosity and gene duplications), multigene family (*Apanius et al., 1997*; *Klein, Sato & Nikolaidis, 2007*) plays a decisive role in controlling the vertebrate adaptive immune system by presenting self- and pathogen-derived peptides to T-cells (*Janeway et al., 2001*). Pathogen-mediated selection is acknowledged to be one of the primary factors of balancing selection maintaining the extreme MHC polymorphism in a population (*Eizaguirre et al., 2012a*; *Piertney & Oliver, 2006*; *Spurgin & Richardson, 2012*). Therefore, investigating shifts in MHC allele frequencies (*Eizaguirre et al., 2012b*) may be a particularly informative tool as an indirect way to detect the emergence of diseases (*Sommer, 2005*). Similarly, examining how MHC genetic diversity fluctuates in parallel with the incidence of diseases or parasites can provide indirect evidence for the impact of those selective agents on the dynamics of the host population (*McCallum, 2008*). Lastly, MHC genes have also shown to be informative of demographic events, particularly when selection plays an important role in population reductions (*Sutton et al., 2011*).

The European eel is a highly migratory, semelparous fish whose spawning grounds are located in the Sargasso Sea and whose foraging grounds cover coastal, mixohaline and freshwater habitats (*Harrod et al., 2005*) across much of Europe and even extend to North Africa and the Levantine coast. The post-hatching early larval transport is first facilitated by local currents in the Sargasso sea, connecting the spawning area with the Gulf Stream (*Baltazar-Soares et al., 2014*), and by the North Atlantic gyre, that completes the larval migration to the continental shelf (*Bonhommeau, Chassot & Rivot, 2008*; *Kettle, Bakker & Haines, 2008b*; *Munk et al., 2010*). Once larval eels enter shelf waters, they undergo a series of substantial changes in morphology and physiology: individuals become glass eels, with a fusiform, transparent body that facilitates active swimming towards coastal waters (*Miller, 2009*). The duration of the continental life stage varies on the location of growth habitats, and may last from as little as two years (*EIFAAC/ICES, 2013*) to several decades prior to metamorphose into silver eel. At this stage, energetic reserves are collected to allow sexual maturation and the long spawning migration back to the Sargasso Sea (*Tesch, 2003*).

Although the number of glass eels arriving at continental coasts across Europe experienced a first drop in the 1960s, the major recruitment collapse occurred at the beginning of the 1980s (*EIFAAC/ICES, 2011*). For the subsequent three decades, the recruitment of glass eels has remained as low as 1 to 10% of the values prior to the 1980s (*EIFAAC/ICES, 2011*). The low recruitment regime is hypothesized to have resulted from multiple impairing factors including productivity changes in the Sargasso Sea (*Friedland, Miller & Knights, 2007*), habitat degradation including river regulation, pollution and reduced freshwater habitats (*Prigge et al., 2013*; *Robinet & Feunteun, 2002*), changes in oceanic currents (*Baltazar-Soares et al., 2014*; *Kettle, Bakker & Haines, 2008*), introduction and spread of diseases, such as the EVEX (*Van Ginneken et al., 2005*) and the swim bladder parasite *Anguillicola crassus* (*Kirk, 2003*), resulting in a severe lack of spawners (*Dekker, 2003*). Experimental studies focusing on this invasive nematode for instance revealed that the European eel is unable to mount an effective immune response, thus becoming particularly susceptible to infection (*Knopf, 2006*).

The European eel is currently considered a single panmictic population (*Als et al., 2011*; *Pujolar et al., 2014*), even though punctual deviations from panmixia have been reported (*Baltazar-Soares et al., 2014*; *Dannewitz et al., 2005*; *Wirth & Bernatchez, 2001*). While the vast majority of genetic studies have focused on solving the population structure of the species, few have tried to determine the impact of the recruitment decline on the species' genetic diversity (*Pujolar et al., 2011*; *Pujolar et al., 2013*; *Wirth & Bernatchez, 2003*). Studies that aimed at doing so have evaluated neutrally evolving genetic markers. In 2003, Wirth and Bernatchez (*Wirth & Bernatchez, 2003*) analyzed seven microsatellite loci in 611 European eel individuals and reported no measurable signature of the 1980s recruitment decline. In 2011, Pujolar and coworkers (*Pujolar et al., 2011*) analyzed the diversity of 22 microsatellite markers on 346 individuals. Again, genetic signature of a population reduction was absent. It was therefore suggested that the 1980s decline, although marked in terms of population biology was not sufficiently extreme to affect the diversity of these polymorphic and neutrally evolving loci (*Pujolar et al., 2011*). Employing a genome-wide reduced representation sequencing technique (RAD)—that identified in its vast majority neutral markers—*Pujolar et al. (2013)* did not detect either evidence for a recent decline of genetic diversity to be associated with the drastic drop in recruitment.

Here, we expand on these studies detailing an extensive evaluation of the current genetic status of the European eel species, which includes (1) screening of both neutral and adaptive markers and (2) a temporal approach directly comparing two distinct generations of eels. A temporal approach is regarded as a key requirement when investigating the signature of demographic events on genetic diversity in a wild population (*Sutton, Robertson & Jamieson, 2015*). Specifically for the eel system, evaluating genetic diversity in two distinct, non-overlapping age cohorts is important since the major component of genetic distribution of nuclear markers in this species seems to relate to different age cohorts (*Dannewitz et al., 2005*).

## MATERIAL AND METHODS

### Study scheme

A total of 683 eels were analyzed in this study, 202 of which corresponded to mature silver eels caught in freshwater while undertaking their spawning migration. The other 481 individuals were glass eels collected from 2009 to 2012 immediately upon their arrival in coastal waters (Fig. S1). Number of individuals per sampling site, year of capture, developmental stage, and geographical locations can be found in Table 1. Note that we mostly sampled in one catchment per country and used countries' acronyms to refer to the locations where the specimens were collected. All samples were included in the analyses of mtDNA and microsatellites. Although the main objective of this work was to evaluate the genetic status of the eel species on a temporal scale, we also analyzed spatial patterns of genetic differentiation for comparison with previously published studies (*Als et al., 2011*; *Maes & Volckaert, 2002*; *Pujolar et al., 2014*; *Wirth & Bernatchez, 2001*).

Amongst the 683 individuals, 327 were sequenced at the exon 2 of the MHC class II B gene. This number was achieved as a compromise between costs of 454 sequencing and the need to provide sample sizes sufficient to provide a robust screen for spatial and temporal patterns of local adaptation. Locations and respective sample sizes of fish screened for the MHC gene are highlighted in Fig. S2. Temporal analyses were performed after dividing the dataset in two distinct age groups: the older "*silver eels*" age group and the younger "*glass eels*" group. The first group included individuals born soon after the drop in recruitment (late 1990s and early 2000s) while the second group consisted of very recently recruited individuals (2009 onwards) (*ICES, 2015*). DNA was extracted from fin clips ("*silver eels*") or tail clips ("*glass eels*") with Qiagen DNeasy Kit© Blood and Tissue kit (Hilden, Germany) following the manufacturer's protocol.

### Neutrally evolving mitochondrial marker
#### *Genetic estimates of diversity, population differentiation and demography*

All 683 glass eels were sequenced using Sanger sequencing for the mitochondrial NADH dehydrogenase 5 (*ND5*) exactly replicating (*Baltazar-Soares et al., 2014*). Haplotype diversity (*Hd*) and nucleotide diversity ($\pi$) were calculated for each sampling location in DnaSP v5 (*Librado & Rozas, 2009*). Genetic structure was estimated using Arlequin v3.5 with 10.000 permutations (*Excoffier & Lischer, 2009*). Moment-based demographic parameters that test for changes in effective population size were calculated for each sampling location in DnaSP v5 under the assumption of mutation-drift equilibrium. Tajima's *D* (*Tajima, 1989*) and raggedness' *r* (*Rogers & Harpending, 1992*) were also calculated in DnaSP v5. Ninety-five percent confidence intervals were estimated through coalescence simulations using 1.000 permutations. We evaluated the nucleotide mismatch pairwise distributions within each geographical location (*Rogers & Harpending, 1992*). These distributions were compared to expected distributions under a constant population size and sudden population expansion (*Librado & Rozas, 2009*).

Baltazar-Soares et al. (2016), *PeerJ*, DOI 10.7717/peerj.1868

**Table 1 GPS coordinates and summary statistics for each sampling location.** The G_prefix stands for "glass eels." AD2010, AD2011 and AD2012 refer to the three cohorts captured in Adour, France, in the years of 2010, 2011 and 2012 respectively. The glass eels of BU (Burrishole, (UK), BNIRL (Bann, Nothern Ireland), VSWE (Viskan, Sweden), TENG (Tees, England), EGER (Erms, Germany), OSPA (Oria, Spain), VFRA (Villaine, France), WENG (Wye, England), TITA (Tuscany, Italy), and NIRL (Carlingford, Northern Ireland) were all captured in 2009. All other samples relate to "silver eels."

| Population | GPS coordinates | *n* | nHap | S | *Hd* | $\pi$ | Tajima-D | *He* | *Ho* | *Ar* | FIS |
|---|---|---|---|---|---|---|---|---|---|---|---|
| G_AD2010 (France) | | 155 | 31 | 30 | 0.818 (0.234–0.859) | 0.00443 | −2.3905[**] (−1.6033–1.9883) | 0.7343 (0.2689) | 0.6177 (0.2440) | 2.9 | 0.1593 (0.1365–0.1760) |
| G_AD2011 (France) | 43°31′48″N. 1°31′28″W | 129 | 35 | 32 | 0.862 (0.2965–0.8641) | 0.00514 | −2.0896[*] (−1.6068–2.1393) | 0.7472 (0.2550) | 0.6243 (0.2378) | 2.94 | 0.1650 (0.1375–0.1838) |
| G_AD2012 (France) | | 121 | 34 | 35 | 0.851 (0.2401-0.8568) | 0.00516 | −2.1804[**] (−1.5977–2.0335) | 0.7480 (0.2488) | 0.6245 (0.2344) | 2.94 | 0.1658 (0.1398–0.1831) |
| LC (Ireland) | 54°50′54″N. 5°48′51″W | 19 | 4 | 5 | 0.7135 (0.1053–0.8714) | 0.00482 | −0.6022 (−1.7188–1.8481) | 0.7356 (0.2725) | 0.5769 (0.2695) | 2.91 | 0.2180 (0.1319–0.2482) |
| BT (Ireland) | 54°45′23″N. 6°27′48″W | 17 | 9 | 11 | 0.8456 (0.3235–0.9044) | 0.0069 | −0.9266 (−1.7987–1.9496) | 0.7314 (0.2624) | 0.6528 (0.2695) | 2.88 | 0.1106 (0.0251–0.1243) |
| Q (Ireland) | 54°22′0″N. 5°40′4″W | 15 | 5 | 6 | 0.6381 (0.1333–0.8667) | 0.0043 | −0.6099 (−1.8159–1.7688) | 0.7315 (0.2504) | 0.5521 (0.2693) | 2.87 | 0.2537 (0.1427–0.2716) |
| BU (N.Ireland) | 53°55′4″N. 9°34′20″W | 16 | 4 | 7 | 0.575 (0.1250–0.8833) | 0.0048 | −0.6099 (−1.6965–1.8617) | 0.7477 (0.2662) | 0.6367 (0.2537) | 2.95 | 0.1527 (0.0546–0.1761) |
| BL (Ireland) | 55°9′15″N. 6°42′5″W | 11 | 5 | 5 | 0.8364 (0.1818-0.9091) | 0.00524 | −0.6099 (−1.7116–1.8376) | 0.7060 (0.3007) | 0.6152 (0.2611) | 2.82 | 0.1350 (−0.0315–0.1674) |
| SLC (Ireland) | 54°32′20″N. 5°42′6″W | 9 | 7 | 7 | 0.8056 (0.2222–0.9167) | 0.00612 | −0.7082 (−1.6775–1.7558) | 0.7461 (0.2605) | 0.6301 (0.2936) | 2.93 | 0.1606 (−0.0584–0.1802) |
| DK (Denmark) | 57°29′N. 10°36′E | 19 | 13 | 16 | 0.9474 (0.2924–0.9006) | 0.00703 | −1.6199 (−1.8612–1.9447) | 0.7547 (0.2183) | 0.6658 (0.2193) | 2.93 | 0.1208 (0.0347–0.1432) |
| LL (Ireland) | 54°49′26″N. 5°47′40″W | 13 | 7 | 6 | 0.8462 (0.1539–0.9103) | 0.00545 | −0.0199 (−1.7276–1.7731) | 0.7169 (0.2788) | 0.5865 (0.2776) | 2.83 | 0.1884 (0.0796–0.1975) |
| SLB (Ireland) | 54°26′39″N. 5°35′20″W | 15 | 6 | 5 | 0.8476 (0.1333-0.8667) | 0.00427 | −0.0723 (−1.6850–1.8912) | 0.7462 (0.2648) | 0.6025 (0.2653) | 2.94 | 0.1989 (0.0797–0.2264) |
| GL (Ireland) | 54°49′55N. 5°48′40W | 14 | 10 | 12 | 0.9341 (0.3846–0.9231) | 0.00844 | −0.8279 (−1.7574–1.8956) | 0.7548 (0.2494) | 0.6549 (0.2428) | 2.96 | 0.1377 (0.0057–0.1587) |
| FI (FInland) | 60°26′N; 26°57′E | 19 | 9 | 9 | 0.883 (0.2982–0.8947) | 0.00661 | −0.3195 (−1.6406–1.7888) | 0.735 (0.2600) | 0.6348 (0.2547) | 2.89 | 0.1397 (0.0587–0.1621) |
| PT (Portugal) | 38°46′N. 9°01′W | 17 | 7 | 8 | 0.8309 (0.2279–0.8971) | 0.00644 | −0.1299 (−1.7057–1.9351) | 0.7581 (0.2414) | 0.6472 (0.2387) | 2.97 | 0.1397 (0.0180–0.1752) |
| Ger (Germany) | 54°17′16N. 10°14′54″E | 17 | 6 | 6 | 0.7794 (0.1177–0.8824) | 0.00536 | 0.2311 (−1.7057–1.9718) | 0.7328 (0.2516) | 0.6041 (0.2586) | 2.88 | 0.1809 (0.0779–0.2012) |
| G_BU (NIreland) | 53°55′4″N. 9°34′20″W | 14 | 6 | 6 | 0.78 (0.1429–0.9011) | 0.00494 | −0.26534 (−1.6705–1.6921) | 0.7224 (0.2741) | 0.6294 (0.272) | 2.85 | 0.1332 (0.0309–0.1508) |
| G_BNIRL (NIreland) | 55°9′15″N. 6°42′5″W | 12 | 8 | 9 | 0.909 (0.3030–0.9242) | 0.00629 | −1.02555 (−1.7551–1.7575) | 0.7224 (0.2741) | 0.6294 (0.2720) | 2.81 | 0.1635 (0.0209–0.2125) |

Baltazar-Soares et al. (2016), *PeerJ*, DOI 10.7717/peerj.1868

**Table 1** (*continued*)

| Population | GPS coordinates | *n* | nHap | *S* | *Hd* | *π* | Tajima-D | *He* | *Ho* | *Ar* | FIS |
|---|---|---|---|---|---|---|---|---|---|---|---|
| G_VSWE (Sweden) | 57°13′30″N. 12°12′20″E | 16 | 8 | 7 | 0.00433 (0.1250–0.8750) | 0.00433 | −0.96247 (−1.7280–1.9375) | 0.7432 (0.2700) | 0.6335 (0.2946) | 2.93 | 0.1518 (0.0618–0.1637) |
| G_TENG (England) | 54°37′21″N. 1°9′23″W | 10 | 7 | 7 | 0.911 (0.0000–0.9111) | 0.00515 | −1.11638 (−1.7118–1.6872) | 0.7402 (0.2584) | 0.6591 (0.2404) | 2.9 | 0.1160 (−0.0810–0.1260) |
| G_EGER (Germany) | 48°35′35″N. 9°14′11″E | 5 | 4 | 3 | 0.9 (0.0000–0.9000) | 0.00395 | −0.17475 (−1.4554–1.6407) | 0.7111 (0.2951) | 0.6000 (0.3086) | 2.82 | *0.1724 (−0.2152–0.1724)* |
| G_OSPA (Spain) | 43°17′25″N. 2°7′55″W | 4 | 3 | 2 | 0.833 (0.0000–1.0000) | 0.00282 | −0.7099 (−0.7968–2.0118) | 0.7265 (0.2894) | 0.6705 (0.3483) | 2.85 | 0.0876 (−0.3333–0.0876) |
| G_VFRA (France) | 47°30′20″N. 2°29′57″W | 3 | 2 | 2 | 0.667 (0.0000–1.0000) | 0.00377 | – | 0.7439 (0.2442) | 0.6970 (0.2800) | 2.88 | 0.0800 (−1.0000–0.0800) |
| G_WENG (England) | 51°36′36″N. 2°39′43″W | 5 | 4 | 4 | 0.9 (0.0000–1.0000) | 0.00565 | 0.27345 (−1.1743–1.6859) | 0.7323 (0.2221) | 0.6455 (0.2686) | 2.83 | 0.1315 (−0.2840–0.1315) |
| G_TITA (Italy) | 43°40′47″N. 10°16′36″E | 4 | 3 | 3 | 0.833 (0.0000–1.0000) | 0.00471 | 0.16766 (−0.8173–2.0118) | 0.7175 (0.3169) | 0.5568 (0.3444) | 2.85 | 0.2519 (−0.2609–0.2519) |
| G_NIRL (NIreland) | 54.07°N. 6.19°W | 3 | 3 | 3 | 1 (0.0000–1.0000) | 0.00563 | – | 0.6869 (0.2787) | 0.6636 (0.3513) | 2.71 | 0.0379 (−0.3398–0.0597) |

**Notes.**

The remaining acronyms have the following meaning:

LC, Larne Lagoon; BT, Bann Toome; Q, Quoile; BU, Burrishole; BL, BannLower; SLC, LoughComber; DK, Denmark; LL, LarneLagoon; SLB, Boretree; GL, Glynn Lagoon; FI, Finland; PT, Portugal; Ger, Germany; *N*, Number of samples and respective summary statistics for each location is also shown; nHap, Number of haplotypes; *S*, Segregation sites; *Hd*, Haplotype diversity; Π, Nucleotide diversity; *He*, Expected heterozygosity; *Ho*, Observed heterozygosity; *Ar*, Rarefied allelic richness; Fis, Inbreeding coefficient.

Values in brackets represent confidence intervals, with the exception of He and Ho, which represents standard deviations.

$^*p < 0,05$.

$^{**}p < 0.001$.

## Neutrally evolving nuclear markers
### Genetic estimates of diversity, differentiation and demography amongst freshwater locations

All samples were genotyped for twenty-two microsatellite loci optimized from previous studies (*Als et al., 2011*; *Pujolar et al., 2009*; *Wielgoss, Wirth & Meyer, 2008*). Amplification took place in four PCR multiplexes of four to six loci each. Specifically: multiplex A— 55 °C annealing temperature—included CT77, CT87, CA55, CA58, CT68, and AJTR-37; multiplex B—55 °C annealing temperature—included CT82, CT76, CT89, CT59, CA80, and CT53; multiplex C—60 °C annealing temperature—included C01, M23, AJTR-45, AJTR27, I14, and O08; multiplex D—60 °C annealing temperature—included AJTR-42, B09, B22, and N13. All reactions were performed in a total volume of 10 µl and followed the QIAGEN© Multiplex PCR kit's recommendations. Genotyping was performed on an ABI©3100 Genetic Analyzer. Alleles were called in GENEMARKER©v. 1.91 (Softgenetics LLC, State College, PA, USA).

Nei's unbiased heterozygosity ($He$), observed heterozygosity ($Ho$) and $F_{IS}$ were calculated for each sampling location in GENETIX (1.000 bootstrap, *Belkir et al., 1999*). Rarefied allelic richness ($Ar$) was calculated for each sampling location in HP-RARE v1.0 (*Kalinowski, 2005*). Genetic structure amongst sampling locations was inferred through pairwise comparisons in Arlequin v3.5 and Bayesian clustering in STRUCTURE v2.3.3 (*Pritchard, Stephens & Donnelly, 2000*). STRUCTURE was run assuming a maximum number of possible groups of $K = 26$, i.e., representing the sum of all spatial and temporal partitions of our sample, with 10,000 MCMC repeats after a 1,000 burn-in while assuming an admixture model with correlated allele frequencies. Three iterations were performed for each $K$.

Genetic signatures of a bottleneck were tested for each location using the tests available in BOTTLENECK (*Cornuet & Luikart, 1996*). These methods are sensitive to recent and severe reductions on effective population size ($N_e$) (*Cornuet & Luikart, 1996*). A two-phase mutation model was assumed with 10% of the loci allowed to evolve through stepwise mutation (*Kimura & Ohta, 1978*). Allele frequency distributions were also calculated for each location.

### Genetic estimates of diversity, differentiation and demography— inter-generation level

In order to compare genetic diversity and demographic histories between the two cohorts and to avoid sampling bias from disproportionate number of samples in the two eel age groups ("*glass eels*" $n = 481$ and "*silver eels*" $n = 202$), we performed 10 rounds of re-sampling of the data without replacement using PopTools (*Hood, 2010*), hereafter referred to as "replicates." Replicates were performed based on 50 individuals. This standardization is critical to validate future comparisons, as it has been long acknowledged that sample size affects the detection of genetic signatures of recent bottlenecks (*Luikart et al., 1998*) and the estimation of effective population size (*Waples & Do, 2010*).

Deviations from Hardy-Weinberg equilibrium (HWE) were calculated for each replicate in Arlequin v3.5 (10,000 permutations). Nei's unbiased heterozygosity ($He$), observed

heterozygosity *(Ho)*, allelic richness *(Ar)* and $F_{IS}$ were calculated and compared between groups of replicates, i.e., "*glass eels*" and "*silver eels,*" with two-sided *t*-tests in FSTAT (1,000 permutations) (*Goudet, 1995*). The distribution of genetic variance between "*glass eels*" and "*silver eels*" was assessed with an analysis of molecular variance (AMOVA, Arlequin v3.5) amongst groups of replicates. Demographic history was inferred using two approaches. First, we evaluated the possible genetic signature of the recent population decline using BOTTLENECK (1,000 iterations) for each age group as previously described. Second, we estimated the effective population size *(Ne)* of each replicate of "*silver eels*" and "*glass eels*" using the linkage-disequilibrium method implemented in NeEstimator V2.01 (*Do et al., 2014*). We utilized $P_{crit} = 0.05$, since lower $P_{crit}$ can overestimate *Ne* (*Waples & Do, 2008*). All estimates were obtained with the composite Burrows method (*Weir, 1990*). The unweighted harmonic mean was calculated for each group according to the following equation: $\hat{N_e} = \frac{j}{\sum_{i=1}^{j}(1/N_{e(i)})}$ where *j* is the number of replicates, *i* is a given replicate and $N_{e(i)}$ is the *Ne* estimate of the *i*th replicate (*Waples & Do, 2010*).

## Adaptive marker: diversity and demography of the MHC

We amplified the exon 2 of the MHC class II gene that encodes for the peptide-binding groove of the molecule following protocols optimized for the European eel (*Bracamonte, Baltazar-Soares & Eizaguirre, 2015*). We used the forward primer AaMHCIIBE2F3 (5′-AGTGYCGTTTCAGYTCCAGMGAYCTG- 3′) and reverse primer AaMHCIIBE2R2 (5′-CTCACYTGRMTWATCCAGTATGG-3′) which allow the amplification of different allelic lineages of the MHC class II *β* genes (*Bracamonte, Baltazar-Soares & Eizaguirre, 2015*). Sequencing was performed on a 454©platform at LGC genomics (Belgium) following (*Stiebens et al., 2013a*; *Stiebens et al., 2013b*). Briefly, two independent reactions were prepared for each individual. After a first PCR of 20 cycles, a reconditioning step (dilution 1:5) was performed, and the template was used for a second PCR of 20 cycles. The reconditioning step combined with independent reactions was shown to significantly decrease the number of PCR artifacts (*Lenz & Becker, 2008*) and facilitate allele call (*Stiebens et al., 2013a*). The second set of PCR was performed using the specific MHC primers extended by the 454 adaptors and a 10 bp individual tag. Allele calling and respective assignment to individuals followed (*Stiebens et al., 2013a*; *Stiebens et al., 2013b*) and primarily relied on matching alleles present in both independent reactions (*Sommer, Courtiol & Mazzoni, 2013*). Genotyping using this method has previously been compared to Sanger sequencing and showed its high accuracy (*Bracamonte, Baltazar-Soares & Eizaguirre, 2015*). Even though variants may stem from different loci, we will refer to them as alleles hereafter.

Individual MHC allele numbers detected from the different paralogs, nucleotide diversity, and individual average nucleotide p-distance (*Eizaguirre et al., 2012a*) were calculated for each sampling location and for each group, i.e., "*silver eels*" and "*glass eels,*" in DnaSP v5 and using custom Perl scripts. MHC allele pools were compared amongst sampling locations and between "*silver eels*" and "*glass eels*" with analyses of similarity (ANOSIM) using Primer v6 (*Clarke, 1993*) following (*Eizaguirre et al., 2011*) 1,000 permutations. Correlation between MHC divergence and neutral structure was

calculated using a Mantel test between pairwise $F_{ST}$ matrices (mtDNA and microsatellites) and pairwise Bray-Curtis similarity matrices (MHC).

Minimum number of recombination events ($Rm$) and estimates of recombination rate ($R$) were calculated in DnaSP v5 (*Hudson & Kaplan, 1985*), as well as the relative ($R/\theta$) contribution of recombination ($R$) and point mutations ($\theta$) in the generation of genetic diversity (*Reusch & Langefors, 2005*). Gene conversion was investigated using $\psi$ that measures the probability of a site to be informative for a conversion event ($\psi > 0$, (*Betran et al., 1997*)) between "*glass eels*" and "*silver eels*," using a sliding window method (window length $= 2$, step size $= 1$) implemented in DnaSP v5.

In order to test for the mode of evolution of the MHC in *A. anguilla*, overall positive selection was estimated with a $Z$-test implemented in MEGA v5 (*Tamura et al., 2011*). We tested for signs of codon-specific positive selection using maximum likelihood site models with CODEML implemented in PAML v4.4 (*Yang, 2007*) and the mixed effects model of evolution (MEME) (*Murrell et al., 2012*) implemented in the Datamonkey web server (*Delport et al., 2010*; *Pond & Frost, 2005*). The maximum likelihood procedures evaluate heterogeneous ratios ($\omega$) among sites by applying different models of codon evolution. Three likelihood-ratio tests of positive selection were performed comparing the models M1a (nearly neutral) *vs* M2a (positive selection), M7 (ß) *vs* M8 (ß $+ \omega$), and M8a (ß $+ \omega = 1$) *vs* M8 (*Yang, 2007*). In the models M2a and M8, positively selected sites are inferred from posterior probabilities calculated by the Bayes inference method (*Yang, Wong & Nielsen, 2005*). We further tested for sites that experienced episodic events of positive selection by using MEME. This model considers that $\omega$ varies between sites (fixed effect), and between branches at a site (random effect) (*Murrell et al., 2012*). The null expectation is that all branches have $\omega < 1$. In short, this model allows each site to have its own selection history, contrary to fixed effect models (as the ones implemented in CODEML) that assume constant selective pressures within a branch (*Murrell et al., 2012*).

To assume selection as the main evolutionary mechanism responsible for changes in genetic diversity, sites under positive selection were concatenated for downstream analyses (Positively Selected Sites, PSS). Theoretically, sites interacting with parasite-derived antigens are expected to be under positive selection while other sites, within the same exon, may evolve differently but still maintain the integrity of the MHC molecules. Therefore, we also concatenated the remaining sites (nPSS) and performed identical analyses as for the PSS. Nucleotide mismatch pairwise distributions were calculated for PSS and nPSS of both "*silver eels*" and "*glass eels*" under the assumptions of a constant population size and sudden expansion using DnaSP v5. The historical profile of MHC genetic diversity was investigated with Bayesian skyline plots (BSP) (*Drummond et al., 2005*) in BEAST v1.8 (*Drummond & Rambaut, 2007*). Although these statistical procedures are often used to infer demographic events based on fluctuations of neutral genetic diversities, they have also been used to estimate the strength of adaptive evolution at the organism level (*Bedford, Cobey & Pascual, 2011*), and of functional genomic regions (*Padhi & Verghese, 2008*).

Because of several expected characteristics of the MHC, including a deviation from a neutral mode of evolution, recombination or gene conversion events (*Spurgin et al., 2011*) and trans-species polymorphism (*Lenz et al., 2013*) that also occurs in this species

(*Bracamonte, Baltazar-Soares & Eizaguirre, 2015*) as well as the existence at least three loci (*Bracamonte, Baltazar-Soares & Eizaguirre, 2015*), we did not attempt to associate the substitution rate to a clock-calibrated evolution. As such, we fixed a molecular clock and assumed three different mutation rates: 0.2, 1 (the default parameter) and 5 substitutions per time unit respectively. The substitution model was chosen in jModeltest (Tamura-Nei: Tn93) (*Darriba et al., 2012*; *Guindon & Gascuel, 2003*) and also inserted as parameter in BEAST's runs. Markov chain run was set to a length of $1 \times 10^8$. The historical profile of MHC diversity was reconstructed for PSS and nPSS of both "*silver eels*" and "*glass eels*." Piecewise constant skyline model allowing for five skyline groups were used in three independent MCMC runs to verify consistence in parameter space. We then compared the marginal probability distributions of several parameters amongst the runs. Lastly, we constructed lineage-through-time plots. These plots reflect accumulation of lineages through time translated for a given dated phylogeny (*Nee, Mooers & Harvey, 1992*).

## RESULTS

### Neutrally evolving mitochondrial DNA
### *Molecular indices, population structure and demography amongst sampling locations*

Analyses of 355 bp of the mtDNA *ND5* in 683 European eels revealed 102 haplotypes including 73 singletons (Data S1). Forty-eight randomly picked singletons were verified by independent extraction, amplification and re-sequencing to eliminate possible risks of sequencing errors. After we eliminated the possibility of sequencing errors (48 out of 48 singletons were verified by independent sequencing) we included all 73 singletons in the subsequent analyses. Amongst sampling locations, haplotype diversity ranged between 0.575 (BU) and 0.934 (GL). Nucleotide diversity ranged between 0.003 (G_SPA) and 0.008 (GL), with an average of 0.005 ($\pm$0.001) amongst sampling locations (Table 1). Pairwise $F_{ST}$ comparisons computed from haplotype frequencies amongst the 26 geographically confined groups revealed 17 significant pairwise differentiations however none passed corrections for multiple tests following the false discovery rate (threshold $p = 0.013$ for 26 tests (*Narum, 2006*)). All results are shown in Table S1.

Tajima's *D* values were negative amongst almost all sampled locations, suggestive of population expansion or of population subdivision (*Tajima, 1989*). The three exceptions were GER (Germany), which belongs to a closed system, the Schwentine river, where recruitment is solely mediated by stocking (*Prigge et al., 2013*), as well as G_TITA (Italy) and G_WENG (England), where the values might reflect artificial stochasticity due to low sample sizes. Mismatch distribution analyses performed at the population level showed the typical pattern of a historical population expansion where the peak differs from zero (*Rogers & Harpending, 1992*) (Fig. 1A).

### *Molecular indices, population structure and demography between generations*

Haplotype diversity and genetic diversity between "*silver eels*" and "*glass eels*" were very similar: $Hd_{\text{silver eels}} = 0.821$, $Hd_{\text{glass eels}} = 0.842$; $\pi_{\text{silver eels}} = 0.0048$, $\pi_{\text{glass eels}} = 0.0049$. No

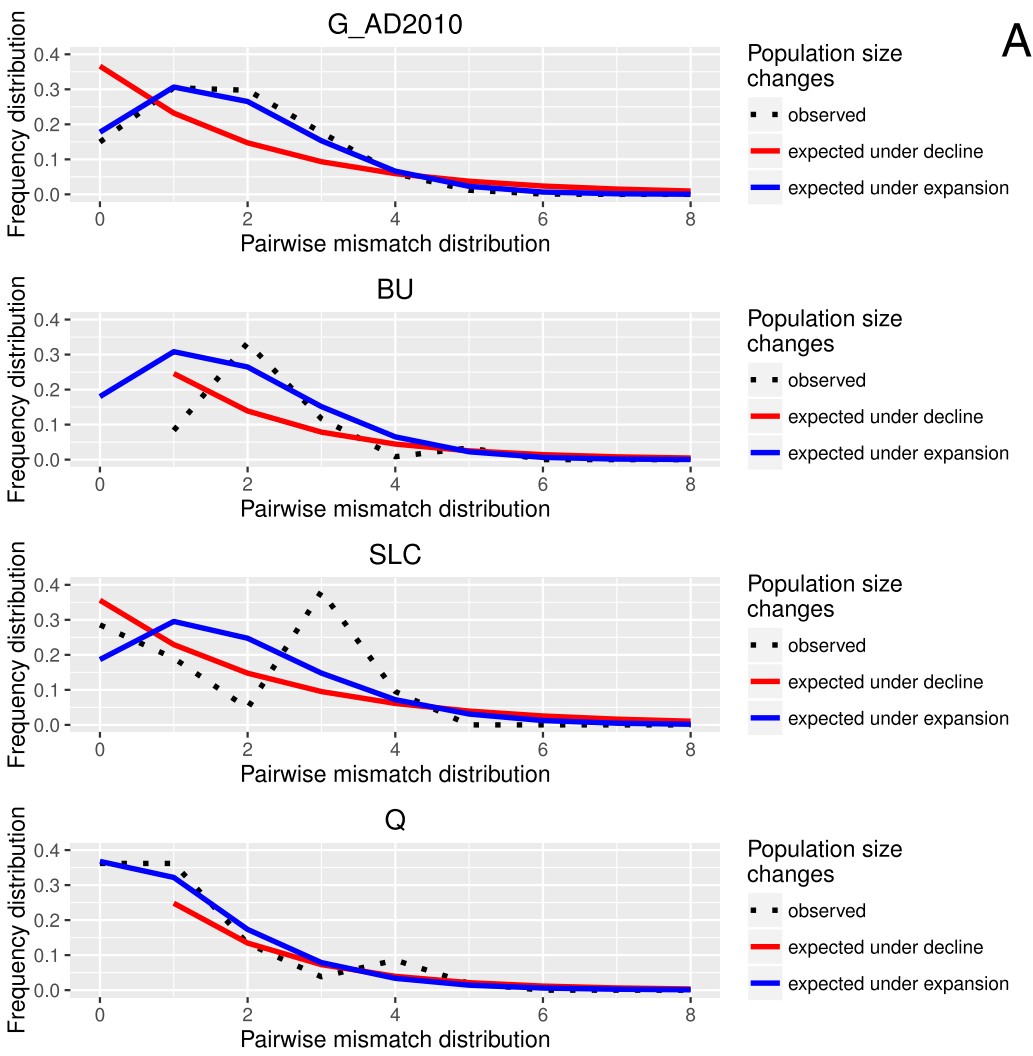

**Figure 1  Location-specific demography accessed with neutral evolving markers.** Two types of analyses specific to each marker are presented in this figure: mismatch pairwise distribution of mtDNA's single nucleotide polymorphisms within locations (A), and frequency distribution of allelic states across microsatellite loci (B). The locations here exposed are G_AD2010 (cohort of 2010 from Adour), BU (Burrishole), SLC (LoughComber), Q (Quoile) and were chosen among the others in order to show the variety of demographic signatures produced by mtDNA data. Therefore in mismatch pairwise distribution graphs, full blue lines represent expected distribution under sudden population expansion, full red lines represent expected distribution under constant population size and dotted lines the observed distribution. The *x*-axis shows the number of mismatches and *y*-axis its frequency. It is possible to observe the signature of an expanding population in G_AD2010, stable population or recovery from bottleneck BU and SLC, and stable population in Q. Allele frequency distribution plots obtained from same locations show the signature of non-bottlenecked population (B). In these graphs, the bars correspond to allele frequencies, *x*-axis corresponds to allele frequency classes and y-plots to number of alleles. 

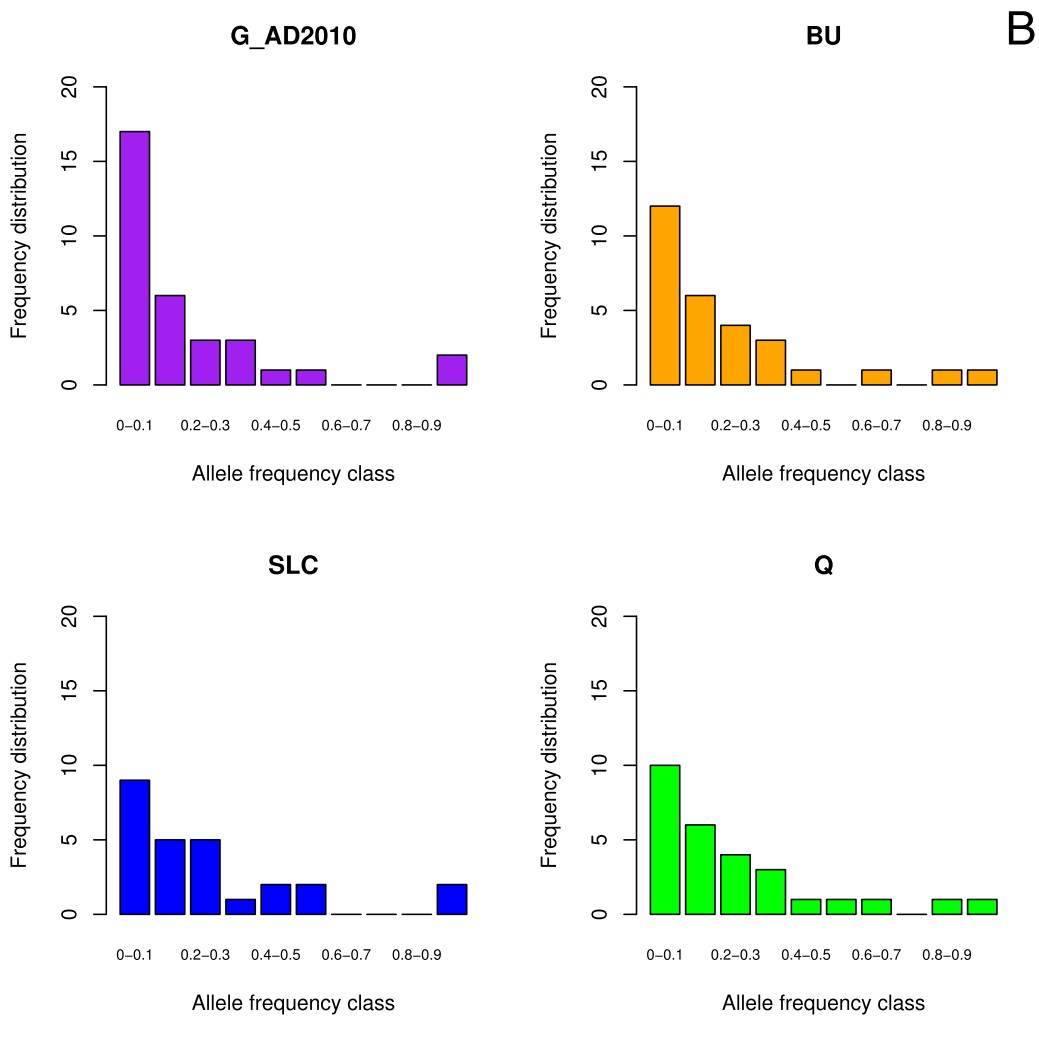

**Figure 1 (…continued)**

evidence for temporal genetic structure based on haplotype frequency distributions was detected between these groups, $F_{ST} = 0.000$, $p = 0.138$. Both groups also had negative and significant Tajima indices: $D_{\text{silver eels}} = -2.053$, $D_{\text{glass eels}} = -2.357$, both $p < 0.05$ (Table 2). Mismatch distribution analyses revealed that both "*silver eels*" and "*glass eels*" display the distribution of expanding populations, suggesting that the overall pattern is not driven by a single generation and has a true biological origin visible in both cohorts (Fig. 2).

### Neutrally evolving nuclear markers
*Molecular indices, population structure and demography amongst locations*
Across populations, *He* ranged between 0.687 (G_NIRL) and 0.758 (PT), *Ho* between 0.557 (G_TITA) and 0.667 (DK) and the average number of alleles per locus varied between 3.546 (G_VFRA) and 15.773 (G_AD2012). Allelic frequencies are reported in Data S2. $F_{IS}$ varied between 0.0380 (G_NIRL) and 0.254 (Q) (Table S1). Even though $F_{ST}$ estimates

**Table 2 Molecular indices of "silver eels" and "glass eels."** Summary statistics of neutral evolving markers calculated for the two distinct generations.

| Population | n | nHap | S | Hd | π | Tajima-D | He | Ho | Ar | FIS |
|---|---|---|---|---|---|---|---|---|---|---|
| Silver eels | 202 | 34 | 33 | 0.821 (0.2230–0.8423) | 0.00481 | −2.05326 (−1.6500–1.9464) | 0.7438 (0.2528) | 0.6233 (0.2376) | 11.74 | 0.1649 (0.1450–0.1802) |
| Glass eels | 481 | 85 | 65 | 0.842 (0.1941–0.843) | 0.00489 | −2.35741 (−1.5842–1.94939) | 0.7427 (0.2598) | 0.6215 (0.2304) | 11.83 | 0.1610 (0.1488–0.1710) |

**Notes.**

$n$, Number of samples used; nHap, Number of haplotypes; $S$, Segregation sites; $Hd$, Haplotype diversity; $\Pi$, Nucleotide diversity; $He$, Expected heterozygosity; $Ho$, Observed heterozygosity; $Ar$, Rarefied allelic richness; Fis, Inbreeding coefficient.

Values in brackets represent confidence intervals, with the exception of He and Ho which represents standard deviations.

\*$p < 0,05$.

\*\*$p < 0.001$.

are very low, pairwise comparisons revealed 7 statistically significant pairwise comparisons after correcting for multiple testing (Table S1). Since six of those included comparisons between locations with low sample size, i .e. G_NIRL, G_WENG, G_TITA or G_BNIRL, the significance could be attributed to stochasticity due to low sample sizes. STRUCTURE analyses did not show any signs of population clustering as expected under the weak observed differentiation (Fig. S3).

None of the sampled locations showed either heterozygote excess or a mode shift in allele frequencies, genetic signatures characteristic of population bottlenecks (Fig. 1B).

### Molecular indices, structure and demography between generations

Twenty-one loci were used to analyze "silver eels" and "glass eels" as locus AjTr-45 consistently deviated from Hardy-Weinberg equilibrium in all "silver eels" replicates. No significant differences between generations were apparent for $He$ ($He_{\text{silver eels}} = 0.744$, $He_{\text{glass eels}} = 0.743$, $p = 0.54$), $Ar$ ($Ar_{\text{silver eels}} = 11.74$, $Ar_{\text{glass eels}} = 11.83$, $p = 0.46$) and $F_{IS}$($F_{IS \text{ silver eels}} = 0.165$, $F_{IS \text{ glass eels}} = 0.162$, $p = 0.07$). $Ho$, however, was significantly higher in the "silver eel" group ($Ho_{\text{silver eels}} = 0.621$, $Ho_{\text{glass eels}} = 0.612$, $p < 0.01$). The AMOVA between "silver eel" and "glass eel" groups of replicates revealed a pattern of isolation by time ($F_{CT} = 0.002$, $p < 0.001$), supporting our a-priori assumption that those groups represent clear age structured cohorts.

None of the replicates showed evidence of heterozygote excess or deficiency. Averaged allele frequencies of neither "silver eels" nor "glass eels" deviated from an expected L-shape distribution (Fig. S3). However, we found that the averaged allele frequencies distribution observed in the "silver eels" group showed the signature of a 5-generations-old bottleneck identified from computer simulations (*Luikart et al., 1998*). This is particularly evident in the distribution of the two most common allele classes, 0.8–0.9 and 0.9–1.0. This signature was not visible anymore in the "glass eel" group (Fig. S4).

Estimates of effective population size ($N_e$) amongst replicates ranged between 0–625.2 for "silver eels" and 0–2708.9 for "glass eels". The harmonic mean of effective population size estimates amongst replicates, $\hat{N}_e$, resulted in $480.9 < \hat{N}_e \text{ silver eels} < 2941.7$ and $1380.2 < \hat{N}_e \text{ glass eels} < 3506.0$ (Table S2).

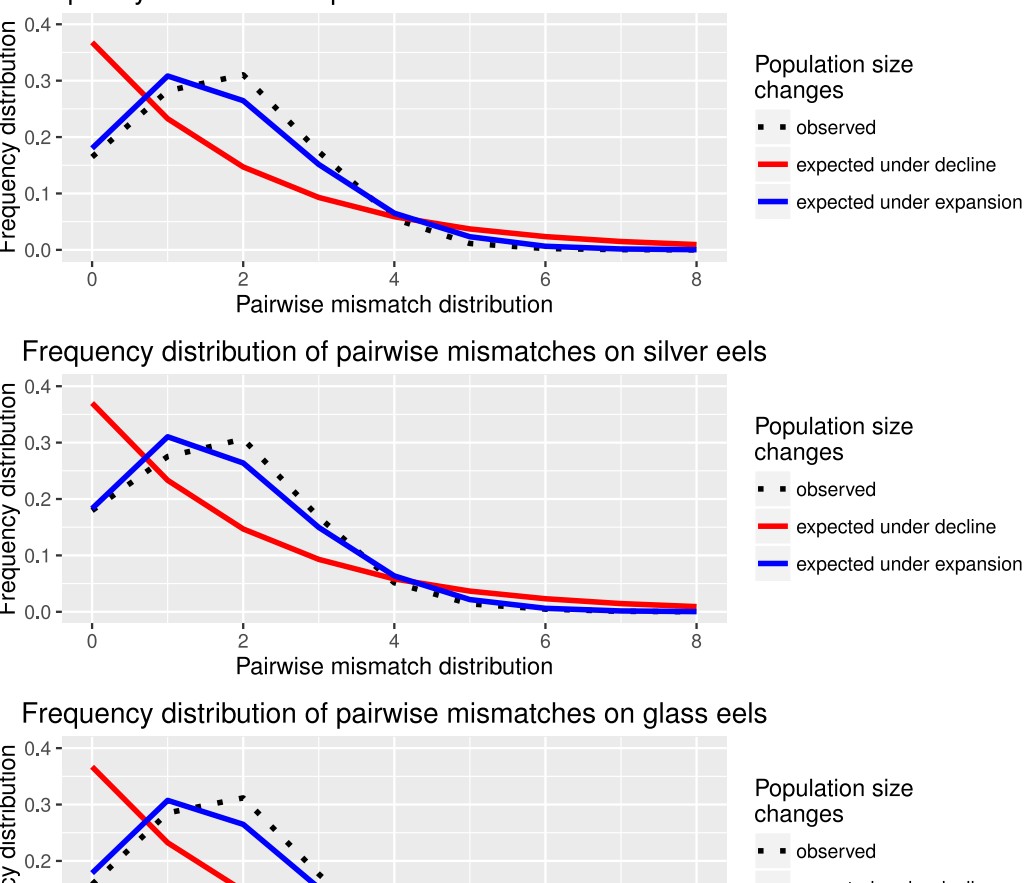

**Figure 2 Species and cohort-based demography accessed with neutral evolving markers.** Mismatch pairwise distribution of mtDNA's single nucleotide polymorphisms considering both the full data set and each generation separately. Here also, the full blue lines represent expected distribution under sudden population expansion, full red lines represent expected distribution under constant population size and dotted lines the observed distribution. All distributions showed a typical signature of expansion.

## Adaptive marker: the MHC
### Molecular indices and population structure

We sequenced a 247 bp fragment of the exon 2 of the MHC class II region (91% of the total size of the exon) in 327 individuals using 454 sequencing technology. We detected 229 different amino acid coding variants. Among those, 226 (98%) were found to be unique in the dataset but present in both independent replicated reactions and therefore kept as true variants (Data S3). A total of 116.276 sequence reads were used in this study. Amongst locations, MHC nucleotide diversity ranged between 0.102 (LL) and 0.138 (BT). The mean number of alleles per individual ranged between 2 (Q, SE = 0.298) and 4 (G_BU, SE = 0.392) (Table S3) and revealed to overall significantly differ amongst sampled

locations ($F_{17} = 1.674$, $p = 0.046$). However, post-hoc pairwise comparisons showed no significant differences between pairs of populations after correction for multiple testing (all $p > 0.05$). The mean nucleotide divergence (p-distance) ranged between 0.078 (BL) and 0.141 (FI) (Table S4) and was significantly different among sampled locations ($F_{17} = 1.860$, $p = 0.021$). Post-hoc pairwise comparisons revealed two significant comparisons after correction for multiple testing (GER *vs* FI, $t = -3.551$, $p = 0.045$, GER *vs* G_AD2011, $t = -3.961$, $p = 0.010$), suggesting a reduced MHC diversity the stocked freshwater system of the Schwentine river, in Germany.

The ANOSIM showed no significant differences in MHC allele pools amongst populations ($R = 0.001$, $p = 0.98$). Overall, no correlation was found between Bray-Curtis similarity matrices on MHC and pairwise $F_{ST}$ for both mtDNA ($R^2 < 0.0001$, $p = 0.58$) and microsatellites ($R^2 < 0.0001$, $p = 0.62$).

Between generations, no difference in MHC allele pools were observed ($R = -0.011$, $p = 0.87$). Interestingly, "*glass eels*" had a significantly higher individual mean number of alleles ("*glass eels*"= 3.423, SE = 0.166; "*silver eels* "= 2.856, SE = 0.101; $F_1 = 8.819$, $p = 0.003$) and a significantly higher individual mean nucleotide p-distance ("*glass eels*" = 0.117, SE = 0.006; "*silver eels*" = 0.101, SE = 0.004; $F_1 = 4.577$, $p = 0.032$) (Table 3). Both the nucleotide diversity ($\pi$) and the number of minimum recombination events ($Rm$) detected between "*silver eels*" and "*glass eels*" were similar ($\pi_{glass eels} = 0.118$, $\pi_{silver eels} = 0.123$; $Rm_{silver eels} = 11$; $Rm_{glass eels} = 10$) while the $R/\theta$ ratio was slightly higher in "*silver eels*" ($R/\theta_{silver eels} = 2.174$; $R/\theta_{glass eels} = 2.089$).

### Screening for novel genetic diversity through events of gene conversion

Gene conversion segments with an average nucleotide length of 4 bp were detected within the "*glass eels*" group but not in the "*silver eel*" group. The average $\psi$ of the whole segment was found to be 0.0002 (Fig. 3). This value is high enough to ascertain the occurrence of conversion events, but not robust enough to determine the exact length of the observed tracts (*Betran et al., 1997*).

### Testing for positive selection

Model-based tests using CODEML revealed 11 sites under positive selection while MEME identified 27 sites that have experienced episodic events of positive selection (Table S4). The discrepancies between the two methods reflect the different assumptions underlying the fixed effect models implemented in CODEML and the mixed effect models of MEME. Positively selected sites detected by both methods matched 10 out of 19 antigen binding sites identified in humans by X-ray crystallography (*Reche & Reinherz, 2003*) (Fig. 3). Due to the functional role of the MHC, all amino acid sites that have experienced at least episodic events of selection were selected for further analyses (Fig. 3)

## Demography and historical profile of MHC genetic diversity

All mismatch distributions indicated a clear deviation from a constant population size, fitting a scenario where a major demographic event occurred (Figs. 4 and 5). The frequency distribution of pairwise differences showed different peaks for both PSS (Fig. 4) and nPSS
**Table 3  MHC molecular indices for "silver eels" and "glass eels."** Summary statistics of the MHC calculated for the two distinct generations.

| Life stage | nAlleles | nIndividuals | nHap | S | h | π | nr alleles/ind | se | dist_nt | se | R | θ | Rm | R/θ |
|---|---|---|---|---|---|---|---|---|---|---|---|---|---|---|
| Glass eels | 332 | 97 | 115 | 100 | 0.9811 | 0.1179 | 3.423 | 0.166 | 0.117 | 0.006 | 48.0000 | 22.9830 | 10 | 2.0885 |
| *Silver eels* | 654 | 230 | 184 | 115 | 0.9820 | 0.1232 | 2.856 | 0.101 | 0.101 | 0.004 | 53.0000 | 24.3850 | 11 | 2.1735 |

**Notes.**

nHap, Number of haplotypes; S, Segregation sites; *Hd*, Haplotype diversity; Π, Nucleotide diversity; *k*, Average number of differences; nr alleles/ind, Average number alleles per individual with respective standard error (se); dist_nt, Average nucleotide distance per individual with respective standard error (se); R, Recombination rate; θ, Mutation rate; *Rm*, Minimum number of recombination events detected; *R/θ*, ratio of recombination and mutation.

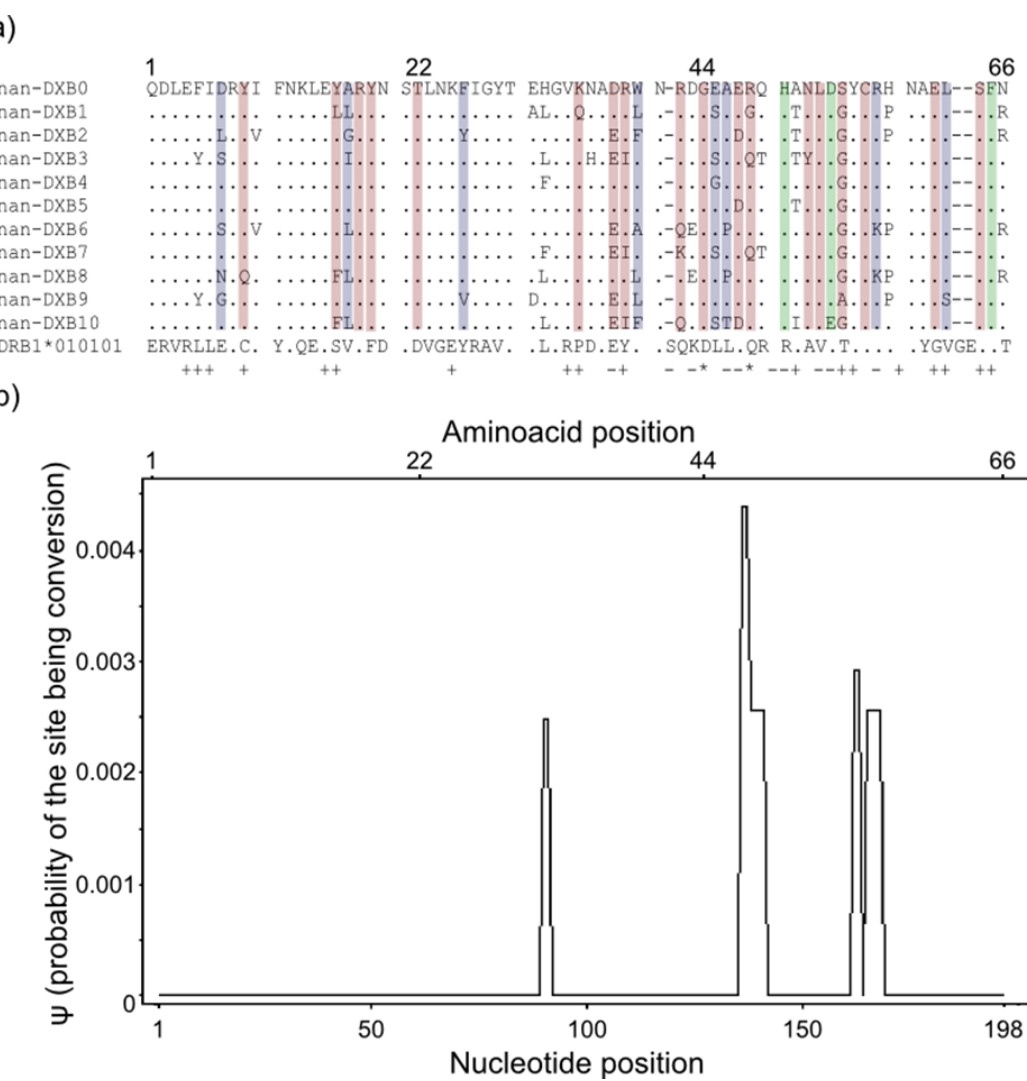

**Figure 3 Landmarks in the allelic sequence of exon 2 of MHC class II B of the European eel.** (A) Aminoacid alignment with human HLA-DRB1. For simplification purposes, only some of the alleles are shown. + denotes human antigen binding sites,—T cell receptor contact sites and * sites that putatively interact with both. Sites estimated to be experiencing or have experienced positive selection are highlighted in green (identified by CODEML only), red (identified by MEME only) and blue (identified by both methods). (B) Sliding window graph of $\psi$. Measures of the probability ($\psi$) of a site being informative of conversion event in relation to the position in the alignment (in base pairs). Here it is possible to observe the two gene conversion tracts detected amongst "glass eels," i.e., the regions 137–141 and 166–168.

(PSS $_{\text{pairwise differences}}$ = 20; nPSS $_{\text{pairwise differences}}$ = 10) (Fig. 5). Those peaks reflect old lineages that are maintained in genes exhibiting trans-species polymorphism as is expected of the MHC (*Klein, Sato & Nikolaidis, 2007*) and recently reported for the European eel (*Bracamonte, Baltazar-Soares & Eizaguirre, 2015*). It was also possible to observe peaks in PSS and nPSS in the frequency of pairwise differences equaling 1. Those peaks are suggestive of increases in genetic diversity. No differences in the demographic profiles were detected between "*silver eels*" and "*glass eels*" (Figs. 4 and 5).

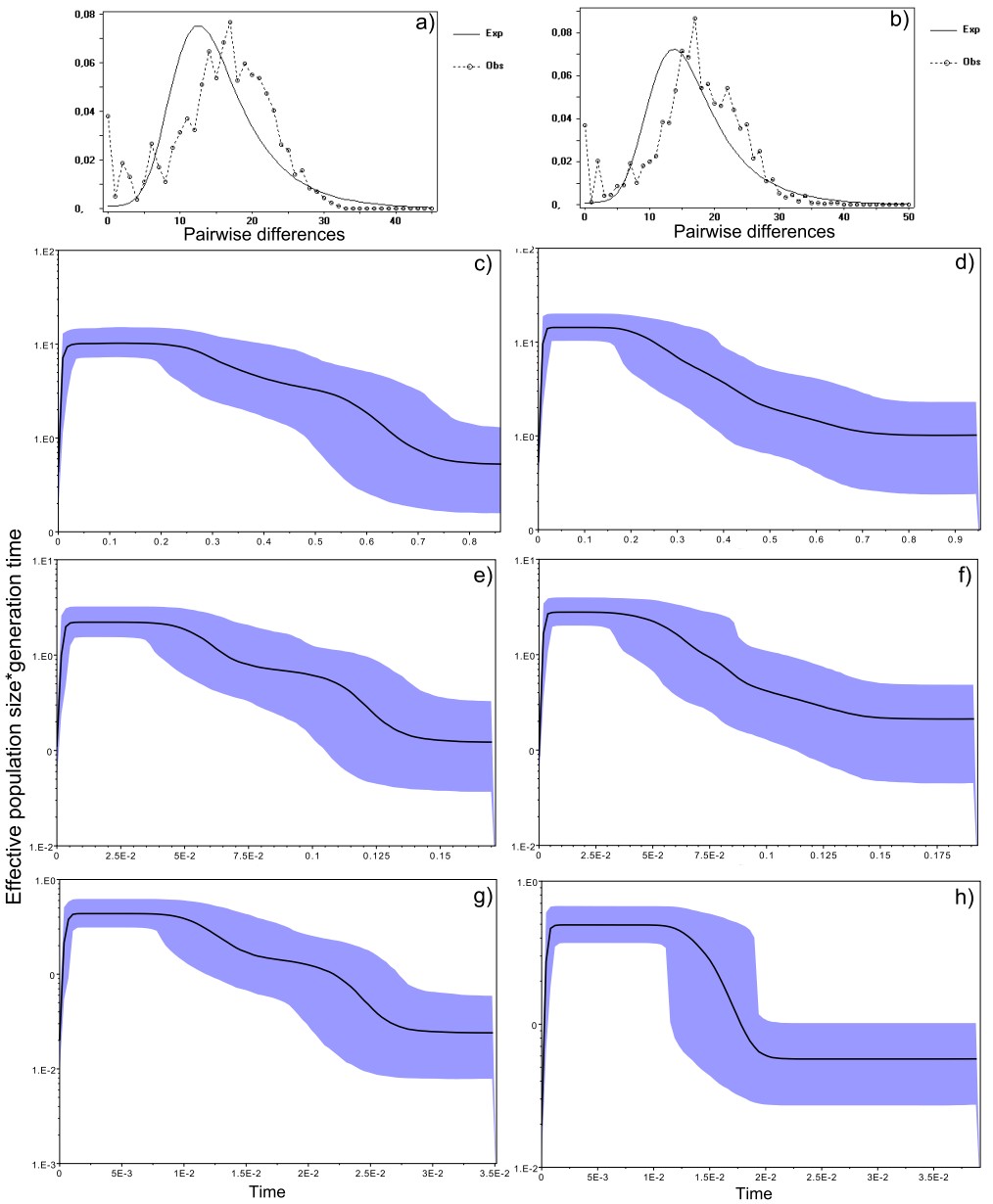

**Figure 4  Historical profile of MHCs genetic diversity considering the positively selected sites (PSS) of the exon 2 of MHC class II B.** Above are the observed mismatch distributions of (A) "glass eels" and (B) "silver eels" (dotted lines) plotted against expected distribution of a population expansion (full lines). Below are the Bayesian skyline plots of "glass eels," considering 0.2 substitutions/ unit of time, (C), 1 substitution/unit of time, (E), and 5 substitutions/unit of time, (G). On the right, Bayesian skyline plots of "silver eels," considering 0.2 substitutions/ unit of time, (D), 1 substitution/unit of time, (F), and 5 substitutions/unit of time, (H). *X*-axis represents "time." The lack of a clock-like evolution did not allow the definition of a time unit. *Y*-axis is an estimate of the product of Ne * mutation rate ($\mu$) per unit of time. The black like represents the mean Ne and the blue shading the 95% HPD (high probability density) interval.

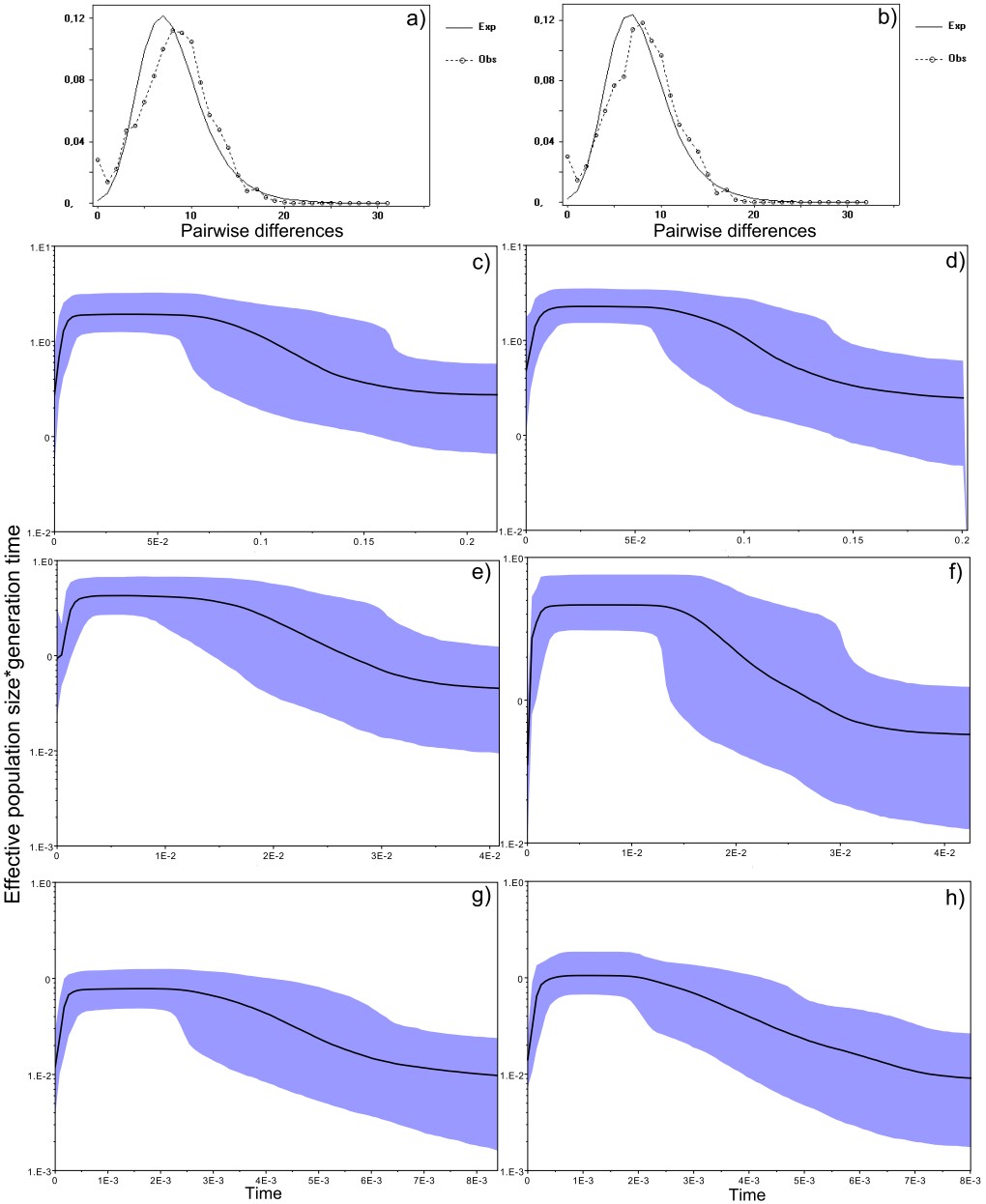

**Figure 5** **Historical profile of MHCs genetic diversity considering the non-positively selected sites (nPSS) of the exon 2 of MHC class II B.** Above are the observed mismatch distributions of (A) "glass eels" and (B) "silver eels" (dotted lines) plotted against expected distribution of a population expansion (full lines). Below, are the Bayesian skyline plots of "glass eels," considering 0.2 substitutions/ unit of time, (C), 1 substitution/unit of time, (E), and 5 substitutions/unit of time, (G). On the right, Bayesian skyline plots of "silver eels," considering 0.0002 substitutions/ unit of time, (D), 1 substitution/unit of time, (F), and 5 substitutions/unit of time, (H).

Bayesian demographic reconstructions revealed a steep decline in genetic diversity that occurred close to the present time. This pattern is characteristic of a genetic bottleneck and is shared by all reconstructions independently of the substitution rates (Fig. 4). However, lineage-through-time plots reveal a very recent burst of lineage diversification, also at $t = 0$

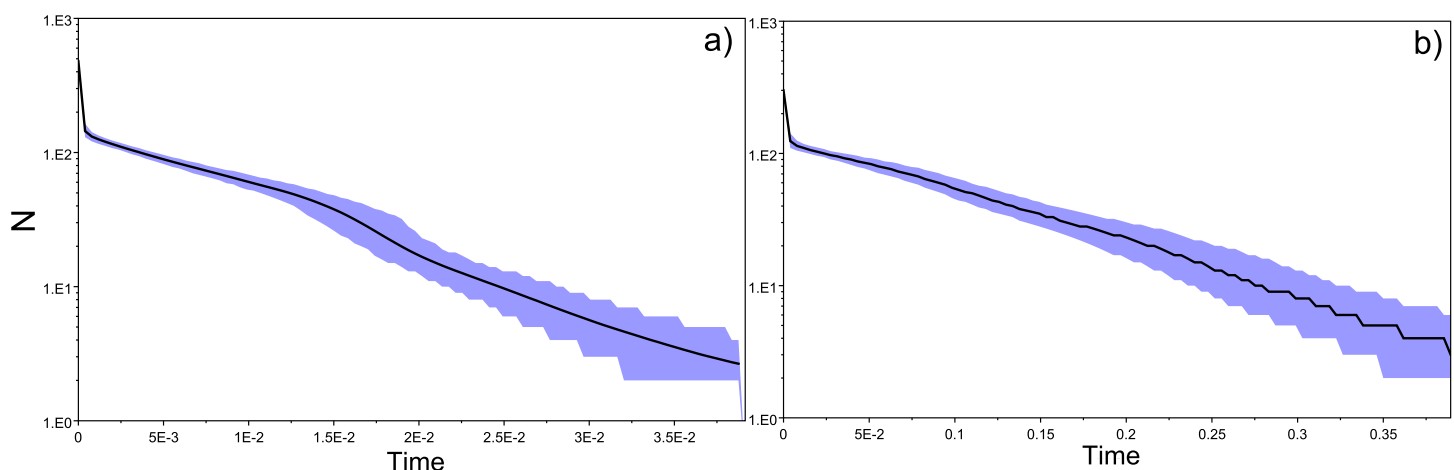

**Figure 6  Lineages-through-time plots (LTT).** Lineage diversification for (A) "silver eels" and (B) "glass eels," with both graphs showing a recent burst of lineage diversification. These graphs were built with PSS using the substitution rate of 5. Like in Bayesian skyline plots, no unit of time is defined. The *y*-axis represents the number of lineages through time.

and common to both generations (Fig. 6). The three independent MCMC runs clearly overlap for the distributions of the posterior, likelihood and skyline estimates (Fig. S5), assuring that the profiles observed were not a product of the Bayesian stochasticity, but rather a real and reproducible pattern.

## DISCUSSION

We investigated how the steep decline in European eel recruitment observed in the 1980s and subsequent population reduction may have affected this species' genetic diversity. We expanded on previous studies (*Pujolar et al., 2011*; *Pujolar et al., 2013*; *Wirth & Bernatchez, 2003*) and searched for contemporary signature of a population reduction by considering neutral markers (mtDNA or microsatellites) as well as a highly polymorphic region of the immune genes of the MHC. Although MHC is an excellent marker to evaluate genetic diversity, as a proxy for adaptive potential, in endangered populations (*Sommer, 2005*; *Sutton, Robertson & Jamieson, 2015*), no formal study of its variation and evolution exist for the European eel.

### Location-specific patterns of genetic diversity and demographic estimates

Although it was not the primary focus of this study, spatial comparison of mtDNA genetic diversity provided information on the genetic differentiation amongst continental locations. Under neutrality, haplotype and nucleotide diversities are predicted to be a function of population size (*Frankham, Briscoe & Ballou, 2002*). Therefore, in the drastically declined European eel population, we expected to find an overall low genetic diversity. Furthermore, and due to reports of panmixia (*Als et al., 2011*), we also expected coherent patterns amongst sampled locations. Instead, we found variation in nucleotide diversity (0.003–0.008), haplotype diversity (0.575–0.934) and Tajima's *D* estimates amongst locations. The high variation in genetic indices amongst geographical areas suggests that processes

act differently across the continental distribution of the European eel. Our observations may result from the post-hatching transatlantic migration, since simulations showed variability in spawning grounds would leave genetic signatures across continental locations under low recruitment (*Baltazar-Soares et al., 2014*). Conversely, it could be explained by a scenario where mtDNA haplotypes are linked to genes under single-generation local selection (*Pujolar et al., 2014*). In this scenario, selection would act mainly in the local foraging environment, and not in the spawning ground, with specific pressures sorting out genotypes in given locations. Expanding the study towards a more genomic approach with adult fish sampled from the spawning ground would reveal further insights into the most prominent scenario.

Due to high levels of polymorphism, neutrally evolving microsatellites are thought to be sensitive enough to detect subtle shifts in population dynamics (*England, Luikart & Waples, 2010*). Here as well, we hypothesized that the chronically low recruitment in European eel observed since the 1980s had negatively affected estimates of genetic diversity in a coherent spatial pattern. Estimates of allelic richness ($Ar = 2.710\text{--}2.960$) and heterozygosity estimates ($He = 0.687\text{--}0.758$) were very similar, and neither mode shifts nor heterozygote excesses were observed (Fig. 1, Fig. S1). These results are in line with a previous study (*Pujolar et al., 2011*), which was conducted focusing on 12 locations (3 locations sampled across a temporal range) and that employed 22 EST-linked microsatellites. The apparent homogeneity of the allelic indices amongst locations matches the expectations based on neutral nuclear markers for a panmictic population where successfully recruited mature fish would mate randomly in the spawning ground (*Als et al., 2011*). Note, after correction for multiple testing, we detected significantly different pairwise $F_{ST}$ estimates amongst some locations. Although this could suggest deviations from panmixia, it is likely this pattern is linked to stochasticity due to low sample sizes in those locations (G_NIRL, G_TITA and G_WENG).

Regarding the MHC, we found differences in the mean number of alleles amongst locations although none of the post-hoc pairwise comparisons revealed to be significant after correction for multiple testing. MHC allele pool composition did not vary amongst sampled locations. We found that the mean nucleotide distances vary amongst locations, but only comparisons between the German populations and two other sampled locations revealed to be significant after correction for multiple testing (Germany *vs* Finland, Germany *vs* France 2011). Whether it relates to the fact that those samples correspond to a freshwater system, the Schwentine in Germany, where all eels are stocked remains to be investigated. Stocking however might be an issue in other sampling areas as well and therefore understanding why this population demonstrates a reduction in diversity should be the focus of further studies.

## Genetic diversity and demographic estimates between age cohorts
### *Recruitment decline of the 1980s did not affected genetic estimates of neutral evolving markers*
The major objective of this study was to compare patterns of temporal variation of genetic diversity post-recruitment collapse observed in the 1980s. Here as well, our mitochondrial DNA results showed (i) no evidence for a genetic bottleneck, (ii) no differences in haplotype

and nucleotide diversities between "*silver eels*" and "*glass eels*," and (iii) no signature of bottleneck in the frequency distribution of pairwise mismatches. The later rather points towards an historical population expansion (*Rogers & Harpending, 1992*). Given the low variability of the mtDNA marker—in comparison with the set of highly polymorphic microsatellites and MHC gene—we suggest that such an event extends back in time to a scenario of expansion related to ice-sheet retreat after the last glacial maxima, as previously proposed for such a pattern (*Jacobsen et al., 2014*).

Investigating the distribution of the genetic variance observed at microsatellites, we found significant differentiation between "*silver eels*" and "*glass eels*" replicates. This pattern of genetic variance distribution is in line with previous reports that also attributed higher genetic variance amongst temporal, rather than spatial, partitions of *A. anguilla* along the European coasts (*Dannewitz et al., 2005*). It also confirmed our a-priori assumption of each group representing a distinct generation, and therefore excludes possible confounding factors associated with overlapping generations from the interpretation of demographic estimates (*Cornuet & Luikart, 1996*; *Waples & Do, 2010*). Note, even though less likely, we cannot exclude that this observed structure also relates to a spatially structured spawning area of this species (*Baltazar-Soares et al., 2014*; *Dannewitz et al., 2005*).

Using microsatellites, we detected no evidence of heterozygote excess in any of the replicates, nor any differences between allelic richness of "*silver eels*" and "*glass eels*." This supports previous reports that the recruitment collapse and subsequent low abundance of the eel population did not leave the expected genetic signatures of reduced genetic diversity (*Pujolar et al., 2011*) suggestive of a system which replenishes genetic diversity rapidly.

However, in-depth demographic analyses suggest that the eel effective population size might actually not be stable. Several lines of evidence support this interpretation: firstly, we estimated $\sim 20\%$ higher effective population size in "*glass eel*" replicates (harmonic mean $N_e = 3506.0$) compared to "*silver eels*" ($N_e = 2941.7$). These contemporary estimates are near the lower confidence intervals of historic effective population sizes previously reported ($5,000 < N_e < 10,000$; (*Wirth & Bernatchez, 2003*), but within contemporary estimates of $3,000 < N_e < 12,000$ (*Pujolar et al., 2011*)). Noteworthy, the confidence intervals calculated in this study for each generation overlap, raising the need to cautiously interpret those results. Secondly, we observed fewer alleles in the most frequent class of allele frequencies in "*silver eels*". The apparent reduction of the most frequent allele class may suggest that the species demography is experiencing a transitory stage from a severe bottleneck, partly detected in "*silver eels*." It is important to mention that such a signature would only be detected under a severe population reduction a few generations in the past. Hence, we could speculate that the hypothetical bottleneck detected only in "*silver eels*" may relate to the drops in European eel recruitment that occurred in the beginning of the 1960s (*EIFAAC/ICES, 2011*). It is possible that the 1960s low recruitment had a major impact on the overall genetic diversity of the species. By the crash in the 1980s, the population would have already been depleted from its original genetic diversity, at least for neutrally evolving markers. Such a scenario requires further studies to be confirmed and would rely on historical samples to be analyzed.

### MHC reveals signatures of selection

Extending the evaluation of genetic diversity to the evolutionary analysis of the adaptive immune genes of the MHC was motivated by two main reasons. The first relates to studies suggesting MHC diversity to be more sensitive than neutrally evolving markers in the detection of demographic shifts (*Sommer, 2005*; *Sutton et al., 2011*). The second relates to the invasion of European freshwater systems by the nematode parasite, *Anguillicola crassus*, for which the MHC was found to respond to in the paratenic host, the three-spined stickleback (*Eizaguirre et al., 2012b*). Using the exon 2 of the MHC class II *β* gene, we evaluated (1) genetic diversity, which might have been affected by the recruitment collapse and subsequent population reduction and (2) allele frequency shifts between generations which would be a signature consistent with directional parasite-mediated selection.

Using next-generation sequencing, we identified a total of 229 MHC alleles amongst 327 individuals. This indicates that the diversity within this species is not low and directly compares to observations made in wild populations of other fishes that are not qualified as endangered, such as for instances, the half-smooth tongue sole (88 MHC class II alleles amongst 160 individuals) (*Du et al., 2011*). Of note, the characterization of the MHC class II genes in this species revealed that up to six different alleles may exist per individual, suggesting the presence of at least three loci (*Bracamonte, Baltazar-Soares & Eizaguirre, 2015*; *Bracamonte, 2013*). Because next generation sequencing is thought to generally overestimate the number of MHC alleles detected (*Babik et al., 2009*; *Lighten, Oosterhout & Bentzen, 2014*; *Sommer, Courtiol & Mazzoni, 2013*), we took multiple precautions to avoid artifacts (reconditioning steps, reduced numbers of PCR cycles, duplicates). Despite such precautions and sequence confirmation using cloning (performed in *Bracamonte, Baltazar-Soares & Eizaguirre, 2015*), we cannot exclude that some variants were called alleles even though artifactual (*Sommer, Courtiol & Mazzoni, 2013*). Again, as mtDNA sequencing revealed a large number of haplotypes ($N = 102$), the large diversity at the MHC displayed in this species may not be surprising.

Generally, our results are suggestive of a pattern of selective sweep at the MHC between the two generations examined here. We found "*silver eels*" to exhibit lower mean number of alleles and lower mean nucleotide distance than "*glass eels*," suggesting that the "*silver eel*" generation was under a selective pressure that reduced its pool of MHC alleles to fewer and more similar alleles. This observation suggests that either the selective pressure was widespread amongst continental locations or that it acted when all eels experienced similar conditions; as for instance, during the fastening spawning migration. A hypothetical selective pressure imposed by *A. crassus* meets both criteria. Not only is this parasite ubiquitous in European freshwater systems (*Wielgoss et al., 2008*) but it also impairs the swimming performance of infected eels (*Palstra et al., 2007*).

Overall, the higher genetic diversity of "*glass eels*" together with the identification of two short gene conversions in this study suggest an ongoing regeneration of the species immune adaptive potential. Indeed, gene conversation is a mechanism capable of generating novel diversity within the MHC region, and seems to be a predominant mechanism in genetically depauperate populations (*Spurgin et al., 2011*).

### Contemporary loss of MHC diversity: evidence of selection?

The parasite *A. crassus* was presumably introduced in the European freshwater systems at the beginning of the 1980s (*Taraschewski et al., 1987*), quickly spreading across continental water bodies. The European eel is particularly susceptible to *A. crassus* infection (*Knopf, 2006*) and therefore its introduction provides an excellent biological calibration to evaluate its impact on the evolution of diversity of the MHC. More specifically, we expected a signature of selection by the parasite to be reflected in positively selected sites of the MHC variants. In total, we detected 27 sites to be under or that have experienced positive selection along their evolutionary history.

Bayesian skyline plots showed a steep decline in MHC genetic diversity as time approaches present. This pattern is visible independently of the substitution rates and is reproducible with independent runs, i.e., overlap of the probability density distributions of the posterior, skyline and likelihood. Together, this suggests a real pattern and not an artifact. This decline in genetic diversity of the adaptive gene is similar to those detected in genealogies exposed to events of episodic positive selection (*Bedford, Cobey & Pascual, 2011*), but also in functional regions involved in adaptive responses (*Padhi & Verghese, 2008*). Two main factors may explain it. Firstly, it can be attributed to the long terminal branching typical of phylogenies of genes evolving under balancing selection (*Richman, 2000*), amongst which the MHC is a classic example (*Klein, Sato & Nikolaidis, 2007*). MHC class II genes are also classical examples of genes evolving through recombination (*Reusch & Langefors, 2005*) and gene conversion (*Spurgin et al., 2011*)—two mechanisms that together with a relatively high copy number variation generates rapid genetic novelty (*Chain et al., 2014*). Therefore, a null expectation for balancing selection and generation of rapid genetic diversity—as predicted for MHC—would rather be that of either an expanding or stable population, which was not what we observed here.

Conversely, it can be attributed to an event of selection speculatively associated with the spread of *A. crassus* across the European freshwater systems. *A. crassus* was unknown to *A. anguilla* before its recent introduction, however it is naturally present in the *A. japonica* population (*Wielgoss et al., 2008*). Hence, the frequency of the MHC alleles, or group of functionally similar MHC alleles, that confer resistance against this parasite would either be low or even absent in the European eel population (*Eizaguirre et al., 2012b*). The selection for those rare variants could have triggered the major loss of diversity we observed in the Bayesian plots and confirmed by the lower diversity indices of the "*silver eels*." Interestingly, the allelic lineage diversity of the MHC showed a constant increase with a particular acceleration approaching the contemporary period, as indicated by the lineages-through-time-plots of both generations. While we are unable to provide a direct functional link between such diversification and our observations of gene conversion and recombination within this specific MHC region, these mechanisms have been associated with signatures of recovery after a genetic bottleneck in genes under balancing selection (*Richman, 2000*). Therefore, the inferred recent steep decline followed by a very recent burst of lineage diversification upholds the occurrence of a selective sweep in the MHC genealogy that pre-dated both our sampling points, as suggested by the comparison between "*silver eels*" and "*glass eels*," with an ongoing recovery of the MHC diversity.

From our results we can hypothesize two scenarios. A first scenario involves a link between a sudden reduction in population size, a loss of genetic diversity and a constant selective pressure extending after the bottleneck. In this scenario, genetic drift would affect overall genetic diversity but since selection would continue to act, genetic diversity of positively selected regions would remain low (*Eimes et al., 2011*). A second scenario relates to multiple MHC loci carrying similar alleles due to recent duplications and to the hypothesis that a population would experience a size reduction and an event of selection within the same time frame. In this scenario, the selective pressure through the bottleneck would lead to a faster fixation of resistant alleles (*Eimes et al., 2011*).

### Limits of the study

Evaluating demography in the European eel is complex particularly due to the difficulty of sampling them at their mating ground and the reliance on indirect genetic evidence. Even though patterns of selection and recent recovery of the MHC diversity seem robust, due to the high allelic variation reported in this gene, we acknowledge that other coalescence models could revealed refined patterns (*Árnason & Halldórsdóttir, 2015*; *Wakeley, 2013*). It is indeed possible that multiple loci analysed together as one locus—although with same functional basis—could create a pattern similar to multiple coalescence. To the best of our knowledge, multiple merger models such as those described by (*Árnason & Halldórsdóttir (2015)* have not yet been applied to investigate the evolutionary signature that linked copies of the same (functional) gene produce on the evaluation of genetic diversity and demography. This thus limits working hypotheses and would be difficult to interpret. Nonetheless, while further investigations are obviously needed to clarify fluctuations in genetic diversity in such a complex but evolutionary relevant immune gene, we argue that our work represents a first empirical step along this line of research.

### Conclusions

In summary, our work reveals signatures of recent reduction in MHC genetic diversity and suggests signs of ongoing recovery of this gene's diversity contributing to the immunogenetic adaptive potential of this endangered species (*Radwan, Biedrzycka & Babik, 2010*; *Sommer, 2005*; *Stiebens et al., 2013b*). Future research will be needed to provide conclusive evidence as to which scenario holds, accommodating also newer theories to further verify the validity of our findings. A future perspective would be to extend the time-series analyses by incorporating screening of MHC diversity in ongoing monitoring practices. This would be a valuable approach to access the evolution of the species adaptive potential.

## ACKNOWLEDGEMENTS

The authors wish to thank J Duhart, K Bodles, D Evans, E Prigge and L Marohn for eel samples; J Klein, L Listmann, J Nickel and M Hoffmann for assistance with laboratory work; M Heckwolf and P Roedler for help in processing glass eel samples; R Scott for help designing the maps.

### Funding

MB-S is funded by the International Max Planck Research School for Evolutionary Biology. CE is partly supported by Deutsche Forschungsgemeinschaft grants (EI 841/4-1 and EI 841/6-1). This work was also supported by a research grant from the Fisheries Society of the British Isles. The funders had no role in study design, data collection and analysis, decision to publish, or preparation of the manuscript.

### Grant Disclosures

The following grant information was disclosed by the authors:
International Max Planck Research School for Evolutionary Biology.
Deutsche Forschungsgemeinschaft: EI 841/4-1, EI 841/6-1.
Fisheries Society of the British Isles.

### Competing Interests

The authors declare there are no competing interests.

### Author Contributions

- Miguel Baltazar-Soares conceived and designed the experiments, performed the experiments, analyzed the data, wrote the paper, prepared figures and/or tables.
- Seraina E. Bracamonte performed the experiments, contributed reagents/materials/ analysis tools, reviewed drafts of the paper.
- Till Bayer and Frédéric J.J. Chain analyzed the data, contributed reagents/materials/ analysis tools, reviewed drafts of the paper.
- Reinhold Hanel and Chris Harrod contributed reagents/materials/analysis tools, reviewed drafts of the paper.
- Christophe Eizaguirre conceived and designed the experiments, performed the experiments, analyzed the data, contributed reagents/materials/analysis tools, wrote the paper.

### Animal Ethics

The following information was supplied relating to ethical approvals (i.e., approving body and any reference numbers):

Our work did not involved experiments with living organisms. Only tissue collected from already dead individuals was used for DNA analyses. All DNA analyses are described in the "Methods" section.

### DNA Deposition

The following information was supplied regarding the deposition of DNA sequences:

Sequences and respective haplotype frequencies for both mitochondrial DNA and MHC are available as Supplemental Information.

## Data Availability

We have provided the complete raw data, the above mentioned sequences and microsatellite allele frequencies as Supplemental Information.

## Supplemental Information

Supplemental information for this article can be found online at http://dx.doi.org/10.7717/peerj.1868#supplemental-information.

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
