# Peer review of "Evaluating the adaptive potential of the European eel: is the immunogenetic status recovering?"

_PeerJ, doi:10.7717/peerj.1868_

## Round 0.1 · original submission · Major Revisions

Two reviewers have assessed your manuscript, and while both of them liked the temporal approach taken in your study, they both also raised some critical issues related to data analysis, interpretation and presentation that will need to be addressed before your paper can be accepted for publication. The main issue raised by both reviewers relates to the MHC sequencing and the treatment of paralogs. Both reviewers raised questions about sequence accuracy using 454 on the MHC as well as whether any other validation has been done in this species for this study. I would also encourage you to more carefully edit your manuscript and ensure that it stands alone (i.e. that the supplementary material is in fact supplementary and not essential to the cohesion of the paper).

Reviewer 1 ·

Basic reporting

This manuscript describes a temporal study on genetic diversity to address the impact of recruitment decline in the European eel, using neutral and adaptive loci in two distinct generations. The authors find no sign of changes in diversity between the generations using neutral markers but they detect spatial genetic differentiation in mtDNA between locations. They find a drop and recent increase in the MHC gene diversity. Temporal studies of this kind are an interesting approach in population genetic studies. Especially in an endangered species such as the European eel which has gone through a drastic reduction in abundance, presumably a bottleneck. Therefore, it is also important to have a selective locus like the MHC, which can
respond quickly to selective factors and show differentiation faster. The research idea is thus sound and much of the analysis work is comprehensive. However, I have several comments that I think the authors have to explain in more detail before publication. The list of my concerns can be found below.

Experimental design

Sound idea but I have some issues about the methods. See the list below.

Validity of the findings

See comments below.

Additional comments

1) Accession numbers for submission of the data to public databases are missing.

2) When amplifying and analysing genes like MHC from a multigene family the difference between orthologous and paralogous genes must be addressed. The authors describe 229 different a.a. variants in a 247 bp long fragment where 98\% are singletons. How can they claim that this is allelic variation of orthologous genes and not variation among paralogous genes? How can mean number of alleles per locus in a diploid organism be 4 (as stated in line number (l.n.) 306)? This must mean that they have included variation of paralogous loci.

3) Expansion of immune genes are known in fish, for example MHC I, Toll like receptors and Cathelicidin genes in Atlantic cod (Star et.al (2011) Nature, 477:207–210; Sundaram et.al (2012) BMC Evolutionary Biology, 12(1):256; K. Halldórsdóttir, E. Árnason Peer J., 3 (2015), p. e976), could the European eel possibly have undergone some similar expansion in numbers of immune genes? Or might “fixed heterozygosity” underlie the the polymorphism found like has been detected in R-genesin plants (Bakker et al. 2006 The Plant Cell Online 18: 1803–1818)?

4) Information of DNA source and extraction is missing.

5) A table with an overview of sample names and populations used in the analysis could be helpful. (Names in Data S3 are not informative).

6) l.n. 117) year of capture, developmental stage can not be found in Table S1.

7) l.n. 131) 2010 onwards while in line 116 2009 - 2012.

8) l.n. 134) How were the mitochondria samples sequenced? 454 as well?

9) Regarding the STRUCTURE analysis, a K=26 will render all classes unique and yield no structure. There is not enough resolution in Fig. S3 to see what you are presenting. Were the results the same for all K? You have to show results for different values of K.

10) l.n. 222) what do you mean by trans-species polymorphism? Here you are analysing only one species.

11) l.n. 240) When dealing with errors of this kind one has to start from scratch. If the source of the error is in the PCR reaction, one cannot count on detecting it by re-sequencing. Did you repeat your PCR reaction when evaluating the singletons?

12) l.n. 345) Those peaks likeley arise ... Please explain better what you mean by this sentence.

13) l.n. 355) How can this scenario that you describe for glass eels be supported by lineages-through-time plots for silver eels in Fig. 5? What is the difference between Fig.4 and Fig.5? Here you show silver eels results and use that to argue about glass eels.

14) Regarding your interpretation of the Bayesian analysis in Figure 4. It is not clear from the Figure that there is a wider high-density probability interval near t=0. Furthermore, the end point in lineage-through-time plot of this nature must always have much less relevance then the rest of the plot and one should be careful not to draw inferences about results at the boundaries (t=0 and t=end) in an analysis of this kind.

15) l.n. 426-428) The meaning here is somewhat contrary to what is stated elsewhere in the manuscript about no genetic differentiation found between the two generations using neutral markers. Please be consistent through the manuscript about this.

16) l.n. 476 ``using cloning'' If you cloned your fragments you need to describe the cloning. Information about that is totally missing from the manuscript.

17) I find it strange that you use 454 next generation sequencing to sequence 247 bp fragment. It seems to be an overkill to use this method for sequencing such a short fragment. Also this technique is prone to error but has the potential to easily detect and cover the whole MHC locus in the genome. Using Sanger sequencing and even cloned fragments, would have give much more accurate sequencing. Some discussion on why this method was chosen would be in order.

18) l.n. 516-518) You have not shown the genetic diversity to be generated with gene conversion and recombination.

19) Figure 2) “silver eels” – (c) and (d) - and “glass eels” – (e) and (f) is this correct?

20) In my opinion the Supplemental Information should be included in the methods chapter.

Reviewer 2 ·

Basic reporting

This is a very dense manuscript with numerous analytical approaches being applied to three distinct datasets for each of silver and glass eels. I found that the Figure captions did not adequately describe the content of the figures, especially Figures 1 and 2, and the reviewer needed to consult the supplemental material extensively to define the many abbreviations used. This should be rectified by including all abbreviations in the figure caption and repeating them in the text. More detailed review of basic presentation below:

o It is unclear what G_AD2010, BU, SLC, and Q are in Figure 1, as these locational abbreviations are not defined in the caption or text but rather in the supplemental data. The author needs to define these in the figure caption and note why these particular locations were chosen.
o In Figure 2, it is not clear what the differences between C/D and E/F are.
o Table 1 alignment is screwed up and is difficult to read. Values do not match the text on line 281, but do match those on the previous page. It is redundant to place all numerical data in the text AND the Table.
o Page numbers would have been very helpful for this review

Experimental design

I have no issues with the experimental design.

Validity of the findings

MHC—226/229 unique sequence variants in 327 individuals. Even for MHC, this seems absurdly high. 2-4 alleles per individual suggests two or more loci were being screened, but there was no discussion of locus structure in this species--I note that these authors have recently published a paper defining the MHC loci in eels, why was this not referred to? As so much of the conclusions are based on the MHC results, I would want to see greater verification of their accuracy and a discussion on how merging alleles from multiple loci would impact the statistical approaches applied.

Additional comments

The most intriguing aspect of the paper is in the potential linkages of the parasite with the patterns of MHC variation. While still highly speculative, it would be useful if the authors could define specific methods that could be applied to validate the proposed relationship between the parasite and selection on the MHC.

---

## Round 0.2 · Major Revisions

As you will see from the reviewer comments, the revisions that have been made to your paper have not been sufficient nor have they been communicated clearly such that the reviewer can see where the changes have been made. I therefore cannot recommend your paper for acceptance in it's current form. I strongly recommend that you pay more attention to detail if you decide to revise and resubmit your paper again, and please be aware that your revised version will require another round of peer review.

Reviewer 1 ·

Basic reporting

See below

Experimental design

See below

Validity of the findings

See below

Additional comments

This is the second time that I review this paper. I still have some comments and
concerns that need to be addressed.

First, the line numbering in the original manuscript, the new manuscript and in
the track-changes file differ and it was hard to follow what line numbering the
authors are referring to in their rebuttal letter, which did in most cases not match
any of the files. The authors also seem to have changed the manuscript without
highlighting some of their changes. Therefore, I do not know if I have a correct
overview of the changes. Thus, I might have missed some of their comments and
explanations.

Regarding comments in the rebuttal letter about multiple genes at the MHCI locus and whether they show trans-species polymorphism and balancing selection: It
is fine to interpret the results in concordance with previous findings but the authors
have to state clearly when they are referring to previous work in the field, which
should have references, and when they are describing and discussing their own results. This is still not always clear.

Figure 2: You state that the figures in panel c (which I guess from panel a to
be results assuming the constant population model) and panel d (from what you
say about b to represent the sudden population expansion model) for the silver eels
that they show similar patterns as for e (assuming constant population model) and f
(sudden population expansion model) for the glass eels. This cannot be the correct
interpretation.
In fact, both panels c and d, which are said to be for silver eels, are identical
and show the same deviation of observed from expected values. Panels e and f,
which are said to be for glass eels, show the same pattern of no difference between
observed and expected. As presented this should be interpreted to mean that the
silver eels (c and d) population deviates under both scenarios (constant population
and sudden expansion) but that the glass eels do not (e and f).

Furthermore, and importantly, it seems to me that panels b, c and d are showing exactly the same figure and panels a, e and f also are showing exactly the same figure. This needs an explanation. How can it be that there are no differences in experimental results?

The text in the figure legend should be corrected, a “ is missing and the sentence
..”towards expansion sudden population expansion” should be rephrased.

From the rebuttal letter: 19) Figure 2) “silver eels” – (c) and (d) - and “glass eels”
– (e) and (f) is this correct?
Your answer: Yes, that is correct. We also replaced “amongst” by “within”, which
is a better descriptor of the analysis. - I cannot find these changes.

Regarding the STRUCTURE analysis; You state in your rebuttal letter that there is
not any substructure at any level. Therefore, Figure S3 does not bring any information to the manuscript and should be excluded.

My comment number 18) l.n. 516-518) You have not shown the genetic diversity to be generated with gene conversion and recombination.
Your answer: We rephrased that section (574-576).
I find in line 605-606 (in the changed file): “Such diversification is consistent
with our observations of gene conversion and recombination within this specific
MHC region.” If you are referring to other previously published works of yours,
references are needed. I do not think the authors have shown evidence of gene
conversion and recombination in this study. The data used in this study are pooled
data of many alleles from several genes. It is not obvious how evidence for recombination and gene conversion can be detected in such data.

Regarding the new chapter “Limits of the study”: If the authors did not do any
analysis based on multiple merger coalescence models in order to, as stated in the
manuscript, evaluate the evolutionary signature of linked copies of genes on the
evaluation of genetic diversity and demography, why are they raising this point?
There exist all kinds of methods that were not applied in this study. As correctly
stated, no research has been done using these methods to address those questions. It would have been an interesting study if the authors had performed such an analysis. However, they did not do any such analysis and speculating about the possible results of such analysis adds nothing to the paper.

---

## Round 0.3 · Minor Revisions

Your paper should be acceptable for publication once you have made final changes based on the reviewer comments. The reviewer still has difficulty reconciling the patterns in Figures 1 & 2, which is partly due to sample size differences, but I also think that these figures are substandard and could be improved. Some confusion may be partly attributable to differences in the axis scales from panel to panel, which I think should be standardized. My reading of these figures is that the observed mismatch distributions of glass and silver eels is almost identical (and therefore almost indistinguishable from the combined data), however the figures illustrating the alternate scenarios are drafted with different x and y axis scales, and this (to me!) is confusing.

Reviewer 1 ·

Basic reporting

see comments for the author

Experimental design

see comments for the author

Validity of the findings

see comments for the author

Additional comments

I still have difficulties regarding Figure 2.

This time the author answer my concern about exactly the same figures for different generations so that their data are not experimental data. Data gained by obtaining samples from nature which are used for all kinds of molecular and statistical analysis are in my view experimental data. If they want to call it observational data that is fine, however, the data represent samples from a population.

I thought the data presented in Figure 2 were from the study and if so the results are peculiar. Figure 1 shows analysis from the mtDNA data based on localities. In panel a expected and observed distribution under sudden population expansion is shown for a glass eel sample (according to the legend for Table 1: The G_ prefix stands for "glass eels".) Panel c, e and g in Figure 1 thus show distribution in silver eel samples from different locations under sudden population expansion. The observations (dotted lines) in panels c, e and g are very different from each other. In Figure 2, however, exactly the same figure is shown for observed data for each generation (glass eels and silver eels) and for the full combined data set. How come? I find it hard to believe that the variation shown among different localities, panel c, e and g in Figure 1 for silver eels disappears when combined with each other and the rest of the silver eel samples, panels c and e in Figure 2. And also that the observed mismatch distribution for silver eels (Figure 2 c and e) are exactly the same as the observed mismatch distribution for the glass eels (Figure 2 d and f). The variation is neither seen when all the glass eel samples are combined with the silver eel samples, that is total data set in panels a and b Figure 2. I understand that Figure 1 shows only fraction of the samples, but it is strange that the variation shown among localities will not be seen as deviation from the glass eel samples in Figure 2.

The Authors state: "It is not surprising for a highly conserved gene such as the mitochondrial ND5 to show exactly the same signature from one generation to another" which is correct but according to the manuscript (line 295 - 295), they analysed 355 bp of the mtDNA ND5 in 683 European eels which revealed 102 haplotypes including 73 singletons all used in the analysis. This is a description of polymorphic data (as can be seen in Figure 1).

The authors have chosen to keep their speculation about multiple merger coalescent in the manuscript without further analysis or explanations. However, they should not cite Kingman's paper from 1982 about multiple merger because it is not mentioned in that paper. The Kingman coalescent allows only two lineages to coalesce at a time.

---

## Round 0.4 · accepted · Accept

Thank you for clarifying the Figures, I believe this addresses the remaining concern and your paper can now be published.